# Expression Analysis of ZPB2a and Its Regulatory Role in Sperm-Binding in Viviparous Teleost Black Rockfish

**DOI:** 10.3390/ijms23169498

**Published:** 2022-08-22

**Authors:** Rui Li, Jiangbo Qu, Dan Huang, Yan He, Jingjing Niu, Jie Qi

**Affiliations:** 1MOE Key Laboratory of Marine Genetics and Breeding, College of Marine Life Sciences, Ocean University of China, 5 Yushan Road, Qingdao 266003, China; 2Key Laboratory of Tropical Aquatic Germplasm of Hainan Province, Sanya Oceanographic Institution, Ocean University of China, Sanya 572000, China

**Keywords:** ZPB2a, black rockfish, oocyte development, sperm-binding

## Abstract

Black rockfish is a viviparous teleost whose sperm could be stored in the female ovary for five months. We previously proposed that zona pellucida (ZP) proteins of black rockfish play a similar sperm-binding role as in mammals. In this study, SsZPB2a and SsZPB2c were identified as the most similar genes with human ZPA, ZPB1 and ZPB2 by Blastp method. Immunohistochemistry showed that ovary-specific SsZPB2a was initially expressed in the cytoplasm of oocytes at stage III. Then it gradually transferred to the region close to the cell membrane and zona pellucida of oocytes at stage IV. The most obvious protein signal was observed at the zona pellucida region of oocytes at stage V. Furthermore, we found that the recombinant prokaryotic proteins rSsZPB2a and rSsZPB2c could bind with the posterior end of sperm head and rSsZPB2a was able to facilitate the sperm survival in vitro. After knocking down *Sszpb2a* in ovarian tissues cultivated in vitro, the expressions of sperm-specific genes were down-regulated (*p* < 0.05). These results illustrated the regulatory role of ZP protein to the sperm in viviparous teleost for the first time, which could advance our understanding about the biological function of ZP proteins in the teleost.

## 1. Introduction

The binding of sperm with an extracellular matrix that surrounds the ovulated ovum is essential for reproductive success in sexually reproducing organisms. This specialized extracellular matrix is composed of several glycoproteins and designated as egg envelope, chorion, vitelline envelope (VE), or zona pellucida (ZP) in all sexually reproducing organisms and in some asexually reproducing postnatal animals [1,2]. A structurally intact ZP plays essential roles in oogenesis and fertilization process in mammals: it mediates species-specific gamete recognition and provides a physical barrier to protect the developing embryo [3].

In most mammals, the ZP family contains either three or four proteins depending on the species: ZP1, ZP2, ZP3 and ZP4 [4,5,6]. These four ZP proteins are also known as ZPB1, ZPA, ZPC and ZPB2, respectively [7]. To improve consistency of presentation throughout the whole text, we only use the later nomenclature system from here on. Four glycoproteins (hZPB1, hZPA, hZPC, and hZPB2) are found in human ZP, whereas there are only three glycoproteins (mZPB1, mZPA and mZPC) reported in mouse due to the pseudogenization of ZPB2 [8]. Researchers usually utilize ZP proteins to characterize their relationship with sperm capacitation and/or acrosomal reaction [9,10]. Some literature shows that ZPB1, ZPC and ZPB2 bind mainly to the head region of capacitated sperm, while ZPA binds to acrosome responsive sperm in humans [11,12]. The functions of three ZP proteins have been elucidated by gene-knockout experiments in mouse. ZPB1 provides more of a support structure function, which is not required for sperm binding or fertilization [13]. Nevertheless, the N-terminal region of human ZPA was shown to play an important role in sperm-egg binding in the mice [14,15,16]. To further study this process, scientists have successfully used transgenic mice as the model composed of different combinations of human ZP proteins [16]. Growing evidence indicates the key roles of ZPA in sperm recognition in mammals [14]. The proteins implicated in the process by which sperm bind to and penetrate zona pellucida have also been identified. From these proteins, sperm-specific SPAM1, also called PH-20, is important for sperm to disperse the oocyte-cumulus and bind to zona pellucida normally in mammals [17,18].

Likewise, teleost ZP proteins also contain several glycoproteins including ZPB, ZPC, ZPD, and ZPAX. These ZP members are the homologous genes with mammal ZP genes, although the number of ZP genes is much more than those of mammal species. This is because that teleost often has multiple copies of the ZP genes, perhaps as a result of ancient polyploidization, gene amplification and mutation [7,19]. In mammals, ZP proteins play two roles: one as structural protein that protect the egg, and the other as sperm binding proteins that mediate the fertilization process. Generally speaking, teleost ZP proteins are considered to be only a structural protein due to lack of acrosome in sperm, and sperm enter the egg through the fertilization pore on the egg surface rather than penetrating the ZP protein [20]. In teleost, most of studies regarding ZP genes were mainly focused on their composition and expression in different tissues [21]. Recent studies have found that ZP proteins can form an amyloid-like structure and act as antifreeze proteins for cold conditions in Antarctica [22], which suggests that the expansion of the ZP family members may result in new functions that are adapted to environmental changes in fish. Although ZP proteins may have evolved new functions, there are no studies on their regulation and effects on sperm in fish.

After sperm enters the ovary for five months, the fertilization process occurs in black rockfish [23,24]. We previously found a significant expansion of the ZP family (especially ZPB subfamilies) in comparison with other oviparous and viviparous fishes; we then proposed that zona pellucida (ZP) proteins of black rockfish play a similar sperm-binding role as in mammals [23]. Pseudo-functionalization of the chorion protein and hatching enzyme (cleaved ZP protein) genes exists in some ovoviviparous fish because the embryo receives more protection from the mother rather than ZP protein [25]. We hypothesized that ZP protein may have other functions beyond its structural role—sperm binding, storage, and viability maintenance. In this study, evolutionary analysis displayed that SsZPB2a was the member with the highest identity to mammals’ ZP proteins. Subsequent incubation experiments between sperm and prokaryotic expressed recombinant ZPB2a protein (rSsZPB2a) indicated that rSsZPB2a could bind to the posterior side of the sperm head and increase the survival of sperm in in vitro. In addition, some sperm-specific genes were down-regulated after knockdown of *zpb2a* gene in the post-mating ovarian tissue cultured in vitro. Our results regarding the molecular characterization and expression analysis of *zpb2a*/ZPB2a, as well as its regulatory effect on sperm in vitro, served as new insights for further studies about ZP proteins in viviparous teleost.

## 2. Results

### 2.1. Identification and Molecular Characteristics of Sszpb2a and Sszpb2c

Using local Blast, Ssc_10013305 and Ssc_10016187 were found to be the most similar genes with the mammal ZPB1, ZPB2 and ZPA, both in terms of full protein length and some previously reported ZP fragments that can bind to spermatozoa (Appendix A). Therefore, we analyzed their phylogenetic relationship with other ZP genes from selected species. The phylogenetic tree suggested that Ssc_10013305 and Ssc_10016187 belong to the ZPB2a and ZPB2c subfamilies, respectively. Hence, they were termed as SsZPB2a and SsZPB2c from here on (Figure 1A). Next, we found that they both have the complete ZP domain (Figure 1B and Appendix A) and the typical trefoil domain (TF domain) immediately upstream of the ZP domain just like other reported ZPB proteins [21]. Multiple ZP sequences alignment displayed that ZP domains were conserved among ZPB subfamily proteins from vertebrate species (Appendix A). There were 10 conserved cysteine residues inside the ZP domain for ZPB2a and ZPB2c in teleost (Appendix A). Furthermore, the spatial expression patterns of *Sszpb2a* and *Sszpb2c* genes in adult black rockfish showed that *Sszpb2a* is specificity detected in the ovary, whereas *Sszpb2c* is mainly expressed in the liver (Figure 1C and Appendix A); this suggests that the two proteins are derived in different ways. Moreover, we found that *Sszpb2a* was always highly expressed in the ovary until the time of fertilization (Figure 1D).

### 2.2. SsZPB2a Was Up-Regulated in the Oocyte with Ovary Development from PRM to PRF

Both rSsZPB2a and rSsZPB2c were successfully expressed in DE3 competent cells (Figure 2A–D), and the expressed proteins were mainly in the inclusion bodies after IPTG induction (Figure 2A–D). The predict rSsZPB2a and rSsZPB2c protein was 43 and 48 kDa, respectively. SDS-PAGE results displayed that purified rSsZPB2a and rSsZPB2c proteins were 48 kDa and 53 kDa in mass, considering the recombinant Thioredoxin protein (rTRX) tag that carries around 5 kDa. In addition, the refolded rSsZPB2a and rSsZPB2c were analyzed by both reducing and non-reducing SDS-PAGE. The results showed that proteins in reducing and non-reducing states did not co-migrate on gel filtration for both rSsZPB2a and rSsZPB2c (Figure 2E), suggesting that refolded rSsZPB2a and rSsZPB2c proteins underwent significant conformational changes and formed disulfide bonds during the refolding process. Taken together, these results indicated that the expressed and purified procedures were correct and purified refolded proteins could meet the requirements of subsequent experiments.

In order to explore the expression profile of ovary-specific SsZPB2a during ovary development at protein level, we prepared the polyclonal antiserum targeted rSsZPB2a. To validate that the antiserum is also specific to endogenous SsZPB2a, two different proteins including rSsZPB2a and ovarian proteins were employed using Western blot assay. The predicted weight for the rSsZPB2a and ovarian proteins is around 48 kDa and 43 kDa, respectively. A band at corresponding size could be visualized in the rSsZPB2a and ovary sample using antiserum (Figure 2F). These results proved that the antiserum is specific to the SsZPB2a protein in black rockfish. Subsequently, we found that the protein level of SsZPB2a was gradually increased in ovary from POM1 to PRF stage using Western blot (Figure 3A), which indicated that SsZPB2a is involved in the oocyte maturation process. Additionally, we detected its subcellular localization in ovary from POM1 to PRF stage using immunohistochemistry. The results showed that the positive signal of SsZPB2a protein began to appear in the cytoplasm of oocyte at stage III in PRM ovary (Figure 3B). With the development of oocyte, the protein location gradually transited to the cytoplasm region near the cell member of oocyte at stage IV from POM1 to POM2 stage (Figure 3B). Then the protein signal was located exclusively in the ZP layer region of the outer cell membrane of oocyte at stage V in POM3 and PRF stage (Figure 3B). There was no signal to be found in the negative control group (Appendix A). This result also indicated that our antiserum specifically targeted the endogenous SsZPB2a protein.

### 2.3. RSsZPB2a Bound to the Sperm and Affected the Viability of Sperm In Vitro

In our previous study, we inferred that ZP protein might bind to sperm in the female ovary after mating stage in black rockfish [23]. In this study, the protein level of SsZPB2a was gradually increased during the ovarian follicle maturation; hence we explored whether rSsZPB2a could bind to sperm in vitro by several methods. Meanwhile, rSsZPB2c and rTRX tag proteins were also performed as the control groups. First, the sperm were incubated with magnetic beads coupled with recombinant prokaryotic protein, and this way better simulated the binding of oocytes with sperm in vivo. As shown in Figure 4, sperm were found at the surface of magnetic beads coupled with rSsZPB2a and rSsZPB2c; however, no sperm were found in the magnetic beads coupled with rTRX. Additionally, more sperm were observed in rSsZPB2a than rSsZPB2c groups (Figure 4).

Next, we would like to investigate the detail location where rSsZPB2a and rSsZPB2c bind to sperm. According to the distribution of green fluorescence, we could confirm that both rSsZPB2a and rSsZPB2c bound at the partial posterior end of spermatozoa head (Figure 5). Likewise, the higher binding activity is found between rSsZPB2a and spermatozoa (Figure 5). Furthermore, Western blot assay proved the positive binding activity of rSsZPB2a rather than rTRX to the sperm in black rockfish (Appendix A). Taken together, all these results indicated that rSsZPB2a and rSsZPB2c could bind to the sperm in vitro.

Given that rSsZPB2a has a strong binding activity with spermatozoa, we would like to confirm whether rSsZPB2a could regulate the viability of sperm in vitro. The sperm were incubated with magnetic beads coupled with rSsZPB2a for 3 h in the DMEM/F12 medium containing 10% FBS. Subsequently, AM/PI staining assay was performed to characterize the survival status of the sperm. Compared to the control group, the sperm survival rate was significantly higher after they were incubated with rSsZPB2a magnetic beads (Figure 6). In addition, sperm motility was significantly better throughout incubation in the experimental group than the control group (Appendix A). Collectively, these results suggested that rSsZPB2a not only bound to sperm but also regulated sperm motility in vitro.

### 2.4. Knockdown of Sszpb2a Destroyed the Integrity of the Oocyte and Down-Regulated the Expression of Genes Associated with Sperm

Since SsZPB2a was involved in the oocyte maturation process, we would like to clarify its relationship with oocyte development before fertilization stage. We used the ovary at PRF stage to perform in vitro culture of ovarian tissue mass as shown in Figure 7A. After being cultured for one day, two different siRNA and appropriate amounts of transfection reagent were added into the medium to knock down endogenous *Sszpb2a*. The ovarian tissues treated with 493-siRNA resulted in nearly 40% diminished expression at the mRNA level compared to the Nc-siRNA group (Figure 7B). Obviously, the integrity of oocyte was affected in the 493-siRNA group. For instance, the whole ZP layer in the Nc-siRNA group was observed, while it was fractured or blurred in the 493-siRNA treated group (Figure 7C). Furthermore, we use rhodamine-labeled WGA to label the ZP layer in the different treatment groups. Fluorescence observation showed that the red fluorescence intensity was significantly lower in the 987-siRNA and 493-siRNA treated groups compared with the control group (Figure 7D), which suggested that the knockdown of *Sszpb2a* caused the disruption of the ZP layer structure. Meanwhile, sperm-specific expressed genes in black rockfish such as *spam1*, *crhbp*, and *prm* were down-regulated in the 493-siRNA treatment group compared with the control group (Figure 7E), suggesting that the number of sperm had decreased with the siRNA treatment. SPAM1 and PRM have been characterized as the marker genes associated with the sperm [23,26]. Additionally, *crhbp* was also a sperm-specifical gene in black rockfish based on tissue transcriptome analysis [23]. Thus, we hypothesized that if an incomplete ZP layer affects sperm attachment, then sperm could not continue to stay the relation with ZP layer, which proceeded to have a negative effect on sperm survival and the expressions of these genes.

## 3. Discussion

ZP is a class of glycoprotein structures distributed on the exterior of the oocyte membrane [27]. It has different names in different species, such as zona pellucida in mammals, choriogenin in fish, and vitelline membrane in birds and amphibians [8]. Although almost all multicellular organisms contain this structure on the outside of their eggs, the sequence encoding ZP vary considerably, which illustrates functional diversity of ZP proteins. In mammals, ZP not only acts as a structural protein to support and protect the oocyte, but also prevents multiple sperm from entering the oocyte during fertilization [28]. Polyspermy could occur when ZPA was knocked out in mouse [29]. In teleost especially some non-model species, the functions of ZP proteins were not well understood due to the restricted gene editing technology. Wang et al. systematically characterized the number distribution of ZP members in vertebrates and conducted their expression profiles using an RNA-seq database in Nile tilapia [21]. In terms of expression, ZP members are mainly distributed in the liver and ovary, which is mainly due to two pathways of ZP synthesis: one is synthesized and secreted from the liver into the blood and then transported to the ovary, and the other is synthesized directly in ovary [21]. In this study, *Sszpb2a* was specifically distributed in ovary using transcriptomic data and Western blot analysis, which was consistent with the expression pattern in adult Nile tilapia, suggesting its considerable role in the ovary development. In addition, *Sszpb2c* was specifically detected in liver, suggesting that it may transport to ovary from liver with ovary development. ZP protein continues to synthesize until the egg matures in mammals. In this study, SsZPB2a was initially located at the cytoplasm of oocyte at stage II in black rockfish by immunohistochemistry analysis. Nevertheless, its protein level was gradually elevated in ovary from PRM to PRF stage. Furthermore, we found that SsZPB2a was synthesized at the cytoplasm of oocyte at stage III and then transited to the inner region near the cellar member of oocyte at stage IV and finally distributed at the ZP layer outside the member of oocyte at stage V. Therefore, the developmental transition from stage II to stage III seems to the main translation period for SsZPB2a protein in oocytes of black rockfish.

The morphology and function of ZP in fish are related to the environment in which the eggs live. Recently, some researchers have discovered a new function of its ZP protein with anti-freezing in an Antarctic fish [22], which is a new functionalization of ZP produced by the adaption for reproduction in the cold environment. The types of reproduction in fish are complex and diverse, including oviparity, ovoviviparity and viviparity. The ZP layer on the exterior of the eggs differs significantly between different reproductive modes [25]. For example, the thickness of inner layer of viviparous black rockfish was thinner than that of oviparous devil stinger, and the value of egg diameter over the thickness of black rockfish was three- to four-fold more than that of devil stinger [25]. Because the fertilization event and early embryonic development take place inside the female body, the process is sheltered by the mother rather than protected by the ZP protein alone. On the other hand, the thickness of ZP layer of black rockfish was thicker than that of viviparous platy fish [25], which may be related to the expansion of the ZP family in the former [23]. In our previous study, we proposed that the sperm are stored in the ZP layer of oocytes; in the present study, we firstly confirmed that SsZPB2a and SsZPB2c, two members of the ZPB subfamily, were able to bind to sperm in vitro, which strongly validated our previous conjecture. In addition, the exact location of ZP protein binding to sperm was affected by the status of the sperm. For humans, ZPB1, ZPC, and ZPB2 bound to the posterior side of the sperm head rather than acrosome region after acrosome reaction [11]. In this study, both rSsZPB2a and rSsZPB2c could bind to the posterior side of black rockfish sperm heads, which prompted us to speculate that the sperm state of black rockfish likely resembles that of the mammalian sperm after acrosome response.

Our results supported that rSsZPB2a not only bound to sperm, but also extended sperm survival time and enhanced sperm motility in vitro. In mouse, the N-terminal of ZPA could bind to sperm and then induced the dissociation by enzymes exiting from the egg cortical granules, resulting in rendering it unable to bind with other sperm to prevent polyspermy [29]. Deletion of ZPA and ZPC resulted in significantly less thickness of the ZP and led to polyspermy [30]. The survival and regulation of sperm is a complicated biological process, which is influenced by multiple factors such as Calcium ion concentration, environmental pH value, and surrounding hormone content, etc. [31,32,33]. ZP is reported to be mainly active in the sperm through the induction of acrosome reaction in mammals [34]. When ZP protein comes into contact with incoming sperm, it immediately induces an ERK signaling cascade within the sperm, which in turn stimulates the acrosome reaction in humans [35,36]. In this study, incubating sperm with rSsZPB2a could significantly maintain sperm motility and delay sperm death. All human ZP proteins can bind to sperm and the difference is that ZPB1, ZPC, and ZPB2 bind to sperm with intact acrosomes, while ZPA binds to sperm with acrosomal reactions [35]. On the contrary, almost all fish lack acrosome [36]. Therefore ZPB2a, which has the highest homology with ZPB2 in mammals, may regulate other aspects of sperm metabolic activity and thus affect their viability and survival in black rockfish. The signaling pathway through which rSsZPB2a regulate sperm remains to be investigated in the future.

## 4. Materials and Methods

### 4.1. Fish and Ovaries Samples

Healthy female and male black rockfish were purchased from PuWan quay market (Qingdao, China); six female fishes were sampled every month from late November to early April and kept in the tanks filled with fresh seawater for at least three days. We named the periods sampled as follows: November, pre-mating stage (PRM); December, post-mating stage 1 (POM1); January, post-mating stage 2 (POM2); February, post-mating stage 3 (POM3); and April, pre-fertilization stage (PRF). All samples were collected after the fish were anesthetized with Ethyl 3-aminobenzoate methanesulfonate (Macklin, Shanghai, China). Ovary samples from six female fishes at each stage were sampled. Half of the ovary tissue was preserved in Bouin’s fixative solution at 4 °C overnight for histology and immunohistochemistry assay. The remaining half of the ovary tissue was frozen immediately in liquid nitrogen and stored at −80 °C until protein extraction.

### 4.2. Phylogenetic Analyses, Domain Prediction of ZPB Genes and Transcript Abundance of Sszpb2a/c Genes

The ZPB subfamily protein sequences from human, chicken, frog, stickleback, medaka, and black rockfish were selected for the phylogenetic analysis. The phylogenetic tree was constructed with NJ method in MEGA 7. The functional domains of SsZPB2a and SsZPB2c were predicted using SMART (http://smart.embl.de/, accessed on 23 March 2022) and displayed using DOG 1.0 software (http://ibs.biocuckoo.org/online.php, accessed on 24 March 2022). Transcript abundances of *Sszpb2a*/c in adult tissues and different embryonic developmental stages were analyzed using previous transcriptome data from our lab [23].

### 4.3. Expression of Recombinant SsZPB2a (rSsZPB2a) and SsZPB2c (rSsZPB2c) and Preparation of Antiserum Targeting rSsZPB2a

The cDNA fragment encoding the mature peptide of SsZPB2a and SsZPB2c were amplified using primer (Appendix A) containing the restriction enzyme sites BamHI, HindIII and XhoI, HindIII, respectively. The target fragments were inserted into the pET-28a (+) vector. Then, the constructed plasmids were verified by sequencing (PsnGene, Qingdao, China) and transformed to BL21 (DE3) cells and induced with IPTG at a final concentration of 0.5 mM at 37 °C for 6 h. The details of expression, purification, and refolding of rSsZPB2a and rSsZPB2c were performed as described previously [37,38]. The refolded proteins were analyzed by both reducing and non-reducing 15% SDS-PAGE and stained with Coomassie brilliant blue R-250. The concentrations of the refolded proteins were determined by BCA Protein Assay Kit (CWBIO, Taizhou, China). The rTRX was expressed using blank pET-28a (+) vector with the above methods and treated as the control for the protein binding assay as other literatures [39,40].

Polyclonal antiserum against rSsZPB2a was raised in mouse following routine methods. Briefly, BALB/c mouse were immunized subcutaneously with 100 μg of purified and refold rSsZPB2a emulsified in Freunds Complete Adjuvant (Beyotime, Shanghai, China), followed by another intraperitoneal injection of the same dose of protein emulsified in Freunds Incomplete Adjuvant (Beyotime, Shanghai, China) after two weeks. One week later, 100 μg of protein was injected to the tail vein of the mouse and repeated once after one week. Blood was collected 3 days after the final injection and stored overnight at 4 °C. Then, antiserum was obtained by centrifuge at 10,000 rpm for 5 min at 4 °C.

### 4.4. Western Blot

The protein of ovaries or sperm was extracted using RIPA lysis buffer (Beyotime, Shanghai, China) according to the manufacturer’s instructions. Then the protein samples were separated on 4–15% Bis–Tris gels and electrophoretically transferred to PVDF membranes. The membranes were blocked with 5% non-fat milk in Tris-buffered saline buffer (TBS) at 4 °C overnight and incubated with anti-SsZPB2a or anti-His tag antibody (CWBIO, Beijing, China) or anti-β-actin antibody (Bioss, Beijing, China) diluted at a ratio of 1:1000 in 5% non-fat milk at room temperature for 3 h. The membranes were washed three times with TBST and incubated with Goat anti mouse antibody (CWBIO, Taizhou, China) diluted at a ratio of 1:2000 in 5% non-fat milk at RT for 1 h. After washed three times with TBST, the membranes were detected with Omni-ECL™Femto Light Chemiluminescence Kit (epizyme, Shanghai, China) according to the manufacturer’s instructions.

### 4.5. Histology and Immunohistochemistry

The ovary samples in Bouin’s fixative solution were dehydrated with a graded alcohol series (70%, 80%, 90%, 95%), then transitioned to 100% ethanol, xylene and embedded by paraffin ultimately. The ovarian blocks were sectioned at 5 μm thickness, deparaffinized in xylene, and stained with hematoxylin and eosin staining kit (Solarbio, Beijing, China). Sections for immunohistochemistry were submerged in PBS after dewaxing. Endogenous peroxidase was extinguished with 3% H_2_O_2_, incubated with Sodium Citrate Antigen Retrieval Solution (Solarbio, Beijing, China) at 95 °C for 15 min to retrieve antigens and then washed with PBS three times at room temperature. Then the sections were incubated in 5% Normal Goat Serum diluted in PBS. Subsequently, the primary antibody (anti-SsZPB2a: 1:150 in 5% Normal Goat Serum) was added and incubated overnight at 4 °C. Subsequently, the sections were washed six times in PBS and incubated with HRP conjugated goat anti-mouse IgG (CWBIO, Taizhou, China) at room temperature for 1 h. The sections were then washed six times in PBS, and antigen–antibody complexes were detected with Metal Enhanced DAB Substrate Kit (Solarbio, Beijing, China) following the manufacturer’s instructions. In the end, the sections were stained with Modified Lillie-Mayer Hematoxylin solution (Solarbio, Beijing, China). Negative groups were treated by replacing the primary antibody with a pre-immune serum to check antibody specificity. The staining results were observed and photographed by a Nikon Eclipse Ti-U microscope (Nikon, Tokyo, Japan).

### 4.6. Conjugation of rSsZPB2a and rSsZPB2c to Magnetic Beads

Ni-IDA MagAgarose beads were developed for the affinity purification of proteins with a His-tag. In this study, His-tagged recombinant protein rSsZPB2a and rSsZPB2c were binding to the Ni-IDA MagAgarose Beads (Biomed, Beijing, China) according to the manufacturer’s instructions; a similar approach has been used in the study of mammalian ZP protein [41]. Briefly, 100 μL of magnetic beads were pipetted into a microcentrifuge tube containing 400 μL of binding buffer (50 mM Tris-HCl, 300 mM NaCl, 0–10 mM imidazole, 0.05% Tritonx-100, pH 8.0) and washed twice with 400 μL of binding buffer. An amount of 400 μL of rSsZPB2a or rSsZPB2c at a concentration of 1 mg/mL was incubated with binding buffer (1:1 *v*/*v*) and then the mixture was incubated with the washed magnetic beads for 30 min at room temperature with orbital agitation. Subsequently, the beads coated with rSsZPB2a or rSsZPB2c were washed twice with wash buffer (50 mM Tris-HCl, 300 mM NaCl, 10–80 mM imidazole, 0.05% Tritonx-100, pH 8.0) to remove non-conjugated proteins, resuspended in PBS and stored at 4 °C until use. Meanwhile, beads with incubation with rTRX was considered as the control group. The rSsZPB2a and rSsZPB2c were eluted by adding 100 μL of elution buffer (50 mM Tris-HCl, 300 mM NaCl, 300–500 mM imidazole, pH 8.0). The eluted protein was separated by SDS-PAGE to make sure that they have been bound on the beads.

### 4.7. Sperm Preparation and Binding Assay between Sperm and Beads Conjugated with rSsZPB2a or rSsZPB2c

Healthy male black rockfish were bought from PuWan quay market (Qingdao, China) during from the natural breeding season (November to December). Fish were anesthetized with Ethyl 3-aminobenzoate methanesulfonate (Macklin, Shanghai, China) before semen collection. For sperm collection, an abdominal incision was made to remove testis. The sperm were gently taken out of the spermatic duct by squeezing and moved into clean dry 1.5 mL cryogenic vials, before being immediately placed on crushed ice until further use. Magnetic beads conjugated with recombinant prokaryotic proteins were washed twice in DMEM-F12 medium containing 10% FBS and co-incubated with centrifuged sperm in 500 μL medium for 3 h at 24 °C. The video recording of sperm movement was obtained using the Nikon Eclipse Ti-U microscope (Nikon, Tokyo, Japan). The magnetic beads were then fixed with 4% PFA and stained with DAPI to detect the candidate sperm on their surface. The beads with no sperm could not be stained by DAPI due to lack of nucleus. A minimum of 10 fields of view were selected for each group, and each field of view contained at least 30 magnetic beads. In addition, Calcein-AM and propidium (Meilunbio^®^, Dalin, China) was used to examined viability of sperm in the medium. The Calcein-AM and propidium were added into the media and their final concentrations were 2 μM and 8 μM, respectively. Calcein-AM, a lipophilic vital dye that rapidly enters viable cells, is converted by intracellular esterases to the membrane-non-permeable polar molecule Calcein, which is retained by cells with intact membranes and produces strong green fluorescence. Propidium iodide (PI), on the other hand, can only enter dead cells and bind to the double helix of DNA to produce red fluorescence [42]. At least 100 sperm at each group were counted, and the percentage of living sperm was calculated as sperm viability. To find out the precise location where rSsZPB2a and rSsZPB2c bind in the sperm, approximately 2 × 10^6^ sperm were incubated with rSsZPB2a or rSsZPB2c coupled with FITC in the same incubation condition as above. The sperm incubated with BSA coupled with FITC were treated as the control group. FITC conjugation was performed using the FITC conjugation kit (Sangon Biotech, Shanghai, China) according to the manufacturer’s instructions. After 3 h, sperm were fixed with 4% PFA and stained with DAPI. All observation and statistics of sperm were carried out under the Nikon Eclipse Ti-U microscope (Nikon, Tokyo, Japan). To further validate the binding between sperm and rSsZPB2a, about 2 × 10^6^ sperm were directly incubated with rSsZPB2a in the same medium at the same cultivated condition. Afterwards, the sperm were washed three times with PBS and then centrifuged at 5000 rpm and stored at −80 °C for protein extraction and Western blot.

### 4.8. Knockdown of SsZPB2a in Ovarian Tissue Cultured In Vitro

Three healthy female black rockfish were sampled in February and their ovaries were carefully sampled after being anaesthetized. Half of the ovary was immersed in PBS and cut off into small blocks. After three minutes of stand, the supernatant was passed through 40 μm cell strainer. Sperm were found in the filtrate by microscopic examination. The other half of the ovary was washed six times in PBS containing 5% Pen-Strep-Nystatin Solution (Gibco, Carlsbad, CA, USA). After that, the tissue block inside the ovary was fully exposed after cutting open the ovarian wall in sterile round petri dishes. Subsequently, the ovarian tissues were cut into 0.3 cm^3^ pieces and placed on a pre-impregnated cellulose acetate film supported by agarose blocks pre-impregnated with L-15 medium and other essential ingredients; the detailed preparation was as previously described [43]. After 24 h incubation, 1 μg of two different siRNA targeting ZPB2a and 2 μL of Lipofectamine 3000 transfection reagents (Thermo Fisher Scientific, Waltham, USA) were added to the medium in each well of a 24-well plate. After 48 h, part of the ovarian tissue masses from the different groups were fixed in Bouin’s solution and the other part were treated directly with liquid nitrogen and placed at −80 °C for RNA extraction.

### 4.9. Statistical Analysis

All experimental data were presented as the mean ± SD (standard deviation) of three independent measurements. Significant differences between means were statistically analyzed using one-way ANOVA followed by Games-Howell or Independent-Samples t-test (SPSS 20.0, IBM, New York, NY, USA) with *p* < 0.05 indicating significant level.

## 5. Conclusions

In summary, two members of the ZPB subfamily in black rockfish were identified to be the most similar homologous genes to mammal ZP genes. SsZPB2a was expressed especially in the ovary and its protein expression level was gradually elevated with ovarian development. In addition, SsZPB2a transferred from cytoplasm to the ZP layer of oocytes from the immature to mature status. Recombinant SsZPB2a and SsZPB2c could both bind to the head region of sperm and SsZPB2a had a stronger binding activity compared to SsZPB2c or control rTRX and BSA. In vitro incubation assay between sperm and ZPB2a proteins showed that rSsZPB2a delayed sperm death. After knocking down the expression of ZPB2a, the integrity of ovarian follicles was destroyed, and the sperm-specific genes were inhibited in the post-mating ovaries. Taken together, these results provided some new insights about biological roles of ZP proteins in teleost.

## Figures and Tables

**Figure 1 ijms-23-09498-f001:**
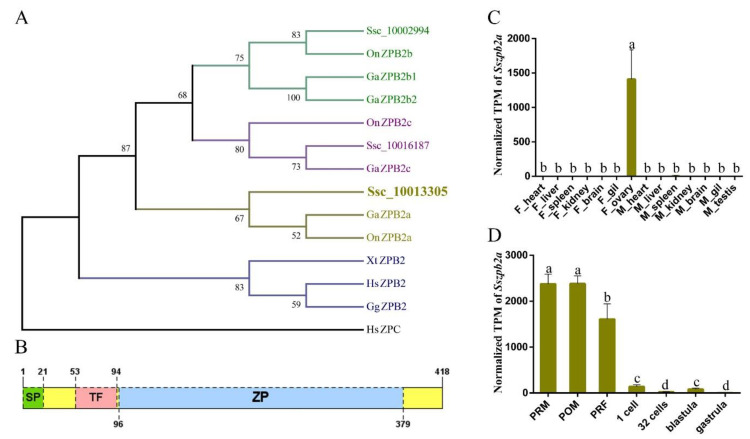
Evolutionary analysis of SsZPB2a and its expression analysis in the adult tissues and embryonic stages at the mRNA level. (**A**) Phylogenetic analysis of the vertebrate ZPB2 family members. Full lengths of the ZPB2 were analyzed by the neighbor-joining method using the MEGA7 program. Bootstrap values derived from 1000 runs are shown. *Homo sapiens* (Hs) ZPC protein is used as an outgroup. Accession numbers for the sequences used are as follows: *Hs* (ZPB2, NP_067009.1; ZPC, NP_001103824.1); *Gallus gallus*, *Gg* (ZPB2, NP_990210.1); *Xenopus tropicalis*, *Xt* (ZPB2, XP_002939919.1); *Oreochromis niloticus*, *On* (ZPB2a, XP_003438185.1, ZPB2b, XP_025764228.1, ZPB2c, XP_025765046.1); *Gasterosteus aculeatus*, *Ga* (ZPB2a, XP_040026904.1, ZPB2b1, XP_040019940.1, ZPB2b2, XP_040019940.1, ZPB2c, ENSGACP00000016351.1). (**B**) Location information of conserved domain of SsZPB2a. (**C**) The expression of *Sszpb2a1* in adult tissues (n = 3). (**D**) The expression of *Sszpb2a1* in embryonic stages (n = 3 for ovaries at each developmental stage, n = 30 for each embryonic stage). Normalized TPM values were used for plotting. TPM, transcripts per kilobase of exon model per million mapped reads. Values with different low letters indicate statistical significance (*p* < 0.05) after one-way ANOVA analysis.

**Figure 2 ijms-23-09498-f002:**
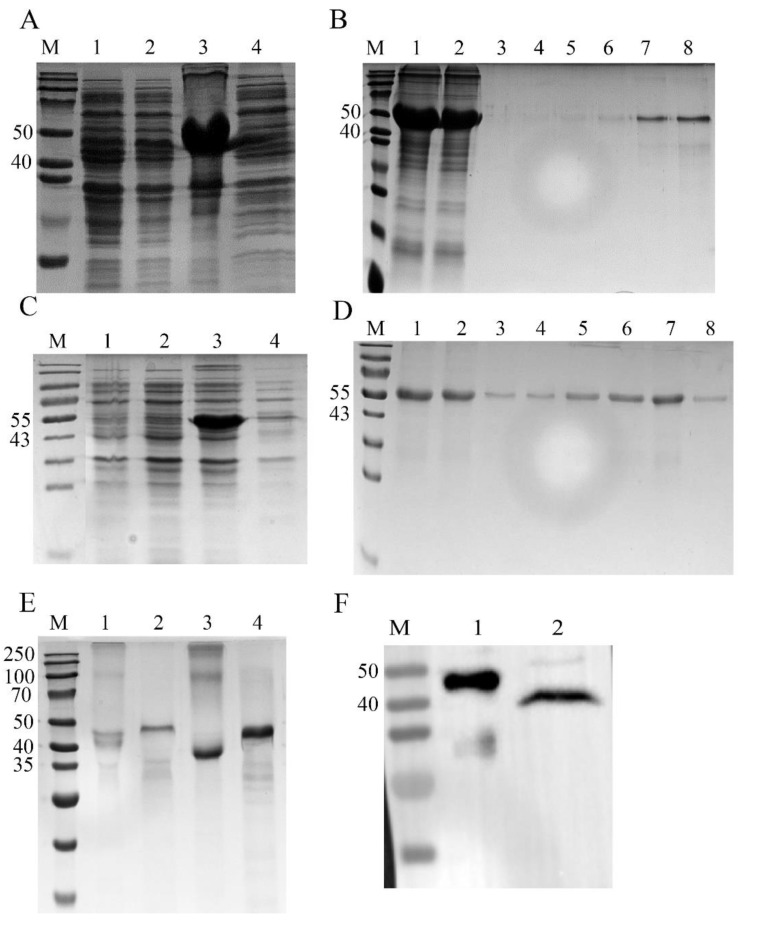
Prokaryotic expression and purification of rSsZPB2a, rSsZPB2c and detection of antiserum targeted SsZPB2a. SDS-PAGE analysis of recombinant protein rSsZPB2a (**A**) and rSsZPB2c (**C**). Lane M: protein marker; lane 1: extracts from expressing strains before IPTG induction; lane 2: extracts from IPTG induced expressing strains; lane 3: the inclusion body protein of extracts after IPTG induction; lane 4: the supernatant protein of extracts after IPTG induction. Purification of rSsZPB2a (**B**) and rSsZPB2c (**D**) analyzed by SDS-PAGE. Lane M: protein marker; lane 1: inclusion body protein prior to binding to Ni-Agarose Resin; lane 2: inclusion body protein after binding to Ni-Agarose Resin; lane 3~8: proteins eluted from different concentrations of imidazole. (**E**) SDS-PAGE analysis of refolded rSsZPB2a and rSsZPB2c. Lane M: protein marker; lane 1: non-reducing SDS-PAGE for rSsZPB2c; lane 2: reducing SDS-PAGE analysis for rSsZPB2c; lane 3: non-reducing SDS-PAGE for rSsZPB2a; lane 4: reducing SDS-PAGE for rSsZPB2a. (**F**) Validation of antiserum of SsZPB2a in black rockfish by Western blot assay. Lane 1: rSsZPB2a protein; lane 2: whole protein extracted from ovary.

**Figure 3 ijms-23-09498-f003:**
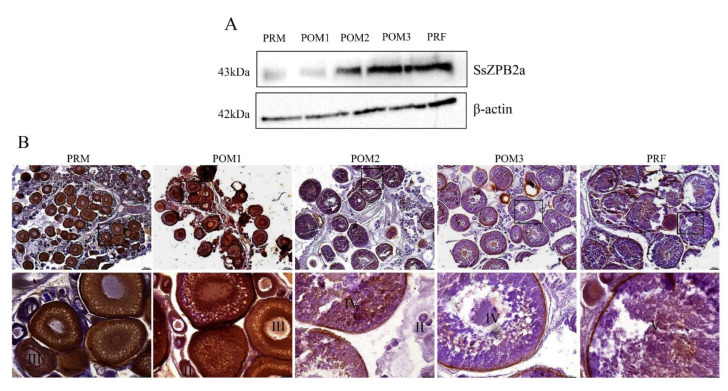
The expression analysis of SsZPB2a in the ovary at different developmental stages at the protein level. (**A**) The expression of SsZPB2a protein during the ovary development (n = 6). (**B**) The immunohistochemistry analysis on the expression of SsZPB2a during ovary development. PRM, pre-mating stage; POM1~POM3, post-mating stage 1~3; PRF, pre-fertilization stage. Scale bar: 200 μm for the low magnification images; 50 μm for the high magnification images.

**Figure 4 ijms-23-09498-f004:**
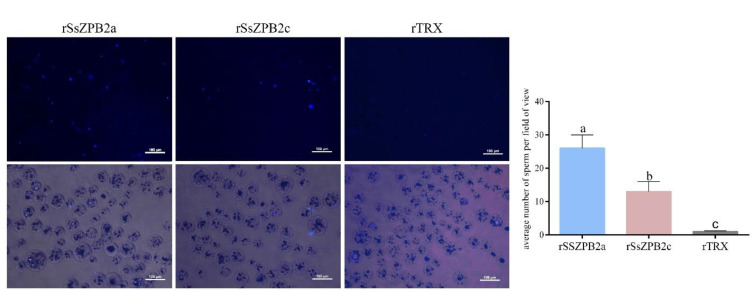
The sperm could bind to the surface of magnetic bead incubated with rSsZPB2a and rSsZPB2c. At least 10 fields of view were selected for each group, and each field of view contained at least 30 magnetic beads. The Data are presented as mean ± SD (n = 10 for each group). Values with different superscripts indicate statistical significance (*p* < 0.05). Scale bar, 100 μm. Values with different low letters indicate statistical significance (*p* < 0.05) after one-way ANOVA analysis.

**Figure 5 ijms-23-09498-f005:**
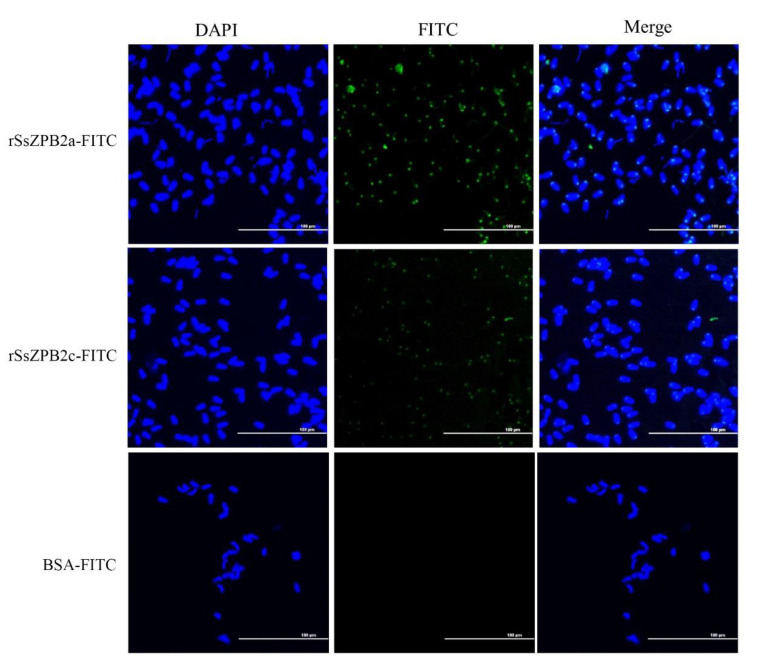
The analysis of the exact location of rSsZPB2a or rSsZPB2c coupled with FITC binding in sperm. Nuclear was staining in blue with DAPI solution. Scale bar, 100 μm.

**Figure 6 ijms-23-09498-f006:**
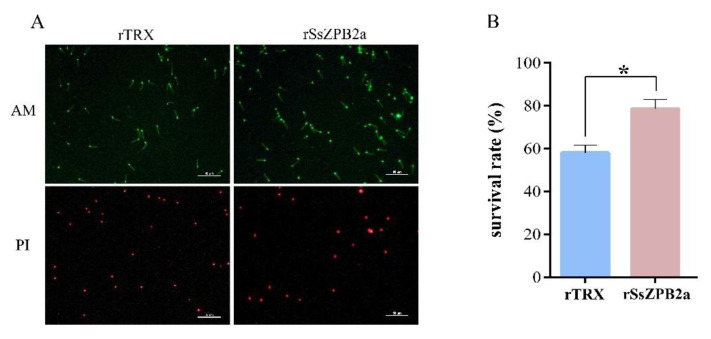
rSsZPB2a significantly enhances the survival rate of sperm cultivated in vitro. (**A**) The AM/PI staining of sperm incubated with rSsZPB2a or rTRX. (**B**) Statistics on sperm survival rate according to the AM/PI staining results. AM, calcein-AM; PI, Propidium iodide. * *p* < 0.05.

**Figure 7 ijms-23-09498-f007:**
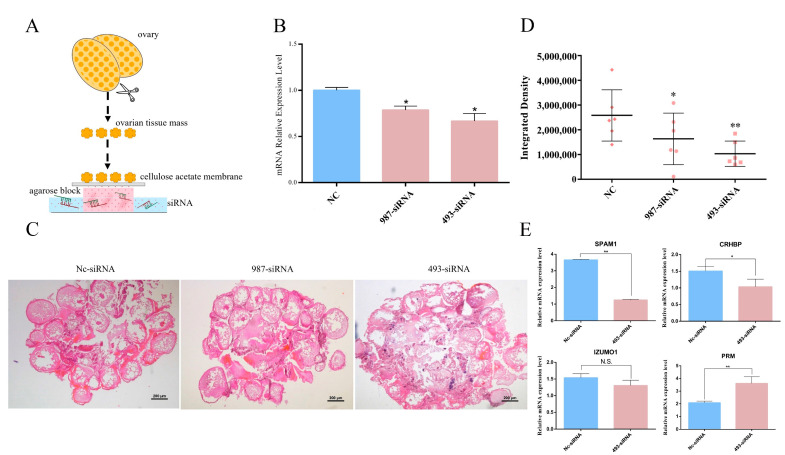
Knockdown of *Sszpb2a* affected oocyte integrity and interfered with the expression of sperm-associated genes in ovarian blocks cultivated in vitro. (**A**) The model of ovarian tissue mass cultivated on pre-impregnated cellulose acetate film supported by agarose blocks pre-impregnated with medium. (**B**) The silencing efficiency of *Sszpb2a* treated with different siRNA in ovarian mass. (**C**) Histological observation of ovarian tissue mass at 48 h after siRNA treatment. (**D**) Fluorescence intensity statistics of the ZP layer staining with WGA coupled with rhodamine in different groups of oocytes. WGA: wheat germ agglutinin. (**E**) The expression of sperm-specific genes in ovarian tissue mass treated with 493-siRNA. * *p* < 0.05, ** *p* < 0.01, N.S., no significant difference.

## Data Availability

The transcriptome data used to analyze the expression of *Sszpb2a* and *Sszpb2c* are available at CNSA (CNGB Nucleotide Sequence Archive) under the accession ID CNP0000222. Zpb2a, zpb2c, zpc3, and zpc1 sequences of black rockfish have been submitted to GenBank with accession numbers ON685210-ON685213.

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
