# Peer review of "Expression Analysis of ZPB2a and Its Regulatory Role in Sperm-Binding in Viviparous Teleost Black Rockfish"

_ijms, 2022, doi:10.3390/ijms23169498_

Round 1
Reviewer 1 Report
The submitted manuscript describes a fairly comprehensive research endeavor that uses a several methods to provide insight into the role of ZPB2a in the black rockfish. The manuscript has several errors in concerning composition using English. Therefore, several errors beyond the few that I will point out in my review need to be addressed. Nevertheless, I think this is an interesting study that provides valuable insight into the role of ZP proteins in reproduction and, following the noted revisions, is a strong scientific publication for the journal.
Revision suggestions presented in chronological order:
Lines
13: The paper switches between using the Spargo-Hope (2003) nomenclature system, for example ZPA, ZPB1, and ZPB2 and the numerical system (like ZP2, ZP1, ZP4). In line 35 you have a statement that can be modified to be a “key” to show that ZP1 = ZPB1, ZP2 = ZPA, ZP3 = ZPC, and ZP4 = ZPB2. If you create a “key” statement then you can improve the clarity and consistency of the paper by using the same nomenclature throughout the paper.
26: The entire first paragraph could use some editing for clarity and improved English composition. This is fairly consistent throughout the paper. For example, the first couple of sentences would read better like: “The binding of sperm with an extracellular matrix that surrounds the ovulated ovum is essential for reproductive success in sexually reproducing organisms.”….
36: ZP4 = ZPB2 and ZP1 = ZPB1
38: Missing citation for pseudogenization statement.
68: “entries” to enters
75: “besides as” to “beyond its structural role “
95: “Besides” can be removed or restructure the sentence
104: Although it is correctly located in the Figure 1 description, Sszpb2a1 added to the y-axis of C and D may provide for better clarity.
117: TPM to be defined. I also did not see any RNA seq information in the materials and methods, please include or explain.
146: “Validation of antiserum of SsZPB2a” Additional information, either in the materials and methods, or results may be needed to ensure that your antibody isn’t cross-reacting with SsZPB2c or any other ZP proteins. It seems that this is a polyclonal antibody, which would likely cause many antibodies to cross-react with common domains shared between SsZPB2a and sSZPB2c proteins as well as the common ZP domain that is present in all ZP proteins. Please clarify how validation to ensure that only SsZPB2a was the lone target of the antiserum and cross-reactivity, either in Western or immunohistochemistry analyses, was limited. This is worrisome since there is only a 5 kDa difference between these two proteins!
168: The Western blot uses beta-actin for loading control. However, you can see through immunohistochemistry, Figure 2B, that SsZPB2a is largely a secreted protein that accumulates in the extracellular matrix. Therefore, I am not sure that I can agree with the statement, as it reads, in line 156 to 157.
177: rTRX is never defined, that I can find, in the entire manuscript.
225: You state that the red fluorescence intensity is “significantly” lower, as depicted in Figure 7D. I can see that there is lower intensity with 493-siRNA, but not 987-siRNA in Figure 7D. Also, I would refrain from using significant since this wasn’t quantified.
Discussion: Several grammatical errors throughout the section
255-256: reword sentence
261: what is ovarian “part”?
265-266: reword sentence for clarity
267: “Consistently” ??
340: the antibody for Western is stated before the description for how it was produced is introduced. I would suggest moving section 4.5 (that describes the production of the antiserum) to after 4.2.
398: “Concentrated” recombinant protein was used to bind to the magnetic beads. Be more specific with quantity of recombinant protein that was used.
429: “At least number of 100 sperm” to “At least 100 sperm”
438: “Furtherly” to further
442: Specify number, if even 1, that was used for ovarian tissue culture
445: Reword sentence
452: “referred to a previous study” to “as previously described”
453: remove the “.” And provide more detail concerning “appropriate amount”
469: “cavity” to activity
472: no need for capitalization of “I” in integrity
Author Response
Response to Reviewer 1 Comments
Comments and Suggestions for Authors
The submitted manuscript describes a fairly comprehensive research endeavor that uses a several methods to provide insight into the role of ZPB2a in the black rockfish. The manuscript has several errors in concerning composition using English. Therefore, several errors beyond the few that I will point out in my review need to be addressed. Nevertheless, I think this is an interesting study that provides valuable insight into the role of ZP proteins in reproduction and, following the noted revisions, is a strong scientific publication for the journal.
Response:Thank you for your review and meaningful suggestions. We have reviewed this manuscript according the reviewers’ requirment. The changed parts have been highlighted with Track change model. The detail point to point response is attatched below, the line number in the revised manuscript (PDF version) indicates the exact location of the change.
Point 1: Line 13: The paper switches between using the Spargo-Hope (2003) nomenclature system, for example ZPA, ZPB1, and ZPB2 and the numerical system (like ZP2, ZP1, ZP4). In line 35 you have a statement that can be modified to be a “key” to show that ZP1 = ZPB1, ZP2 = ZPA, ZP3 = ZPC, and ZP4 = ZPB2. If you create a “key” statement then you can improve the clarity and consistency of the paper by using the same nomenclature throughout the paper.
Response 1: Thank you for your review and suggestion. We believed this advance would be very helpful and meaningful. We have added the statement into the reviesed manuscript (“ZP family contains either three or four proteins depending on the species: ZP1, ZP2, ZP3 and ZP4 [4-6]. These four ZP proteins are also known as ZPB1, ZPA, ZPC and ZPB2, respectively [7]. To improve consistency of presentation throughout whole text, we only use the later nomenclature system from here on.” Line 35-38). And we have changed all numerical system to the Spargo-Hope nomenclature system in the revised text. The revised parts have been marked with “Track Changes”.
Point 2: Line 26: The entire first paragraph could use some editing for clarity and improved English composition. This is fairly consistent throughout the paper. For example, the first couple of sentences would read better like: “The binding of sperm with an extracellular matrix that surrounds the ovulated ovum is essential for reproductive success in sexually reproducing organisms.”….
Response 2: Thank you for your careful review and advance. We have changed rewritten the first couple of sentences as you require (“The binding of sperm with an extracellular matrix that surrounds the ovulated ovum is essential for reproductive success in sexually reproducing organisms. This specialized extracellular matrix is composed of several glycoproteins and designated as egg envelope, chorion, vitelline envelope (VE), or zona pellucida (ZP) in all sexually repro-ducing organisms and in some asexually reproducing postnatal animals [1] [2].” Line 26-30) and modified the English composition over the manuscript with the help of a native English speaker. The modified parts have been marked with “Track Changes”.
Point 3: Line 36: ZP4 = ZPB2 and ZP1 = ZPB1
Response 3: Thank you for your suggestion. We have used the consistent Spargo-Hope nomenclature system in the whole revised manuscript.
Point 4: Line 38: Missing citation for pseudogenization statement.
Response 4: Thank you for your careful review. We have added the citation and reference for pseudogenization statement in the revised manuscript (Line 40). Therefore, we have adjusted all the order of references in the revised manuscript.
Point 5: Line 68: “entries” to enters
Response 5: Thank you for your meaningful comment. We have changed “entries” to “enters” in the revised manuscript (Line 102)
Point 6: Line 75:“besides as” to “beyond its structural role “
Response 6: Thank you for your suggestion. We have changed “besides as” to “beyond its structural role” in the revised manuscript (Line 109).
Point 7: Line 95: “Besides” can be removed or restructure the sentence
Response 7: Thank you for your advance. We have restructured the setence as “Multiple ZP sequences alignment displayed” (Line 156-157).
Point 8: Line 104: Although it is correctly located in the Figure 1 description, Sszpb2a1 added to the y-axis of C and D may provide for better clarity.
Response 8: Thank you for your comment. We have rewritten the y-axis content as “Normalized TPM of Sszpb2a” in the revised Figure 1.
Point 9: Line 117: TPM to be defined. I also did not see any RNA seq information in the materials and methods, please include or explain.
Response 9: Thank you for your suggestion. We have added the definition of TPM into the Figure 1 legend (Line 177-178). And we have added the RNA-seq information into the Materials and Methods 4.2 part (Line 458-460).
Point 10: Line 146: “Validation of antiserum of SsZPB2a” Additional information, either in the materials and methods, or results may be needed to ensure that your antibody isn’t cross-reacting with SsZPB2c or any other ZP proteins. It seems that this is a polyclonal antibody, which would likely cause many antibodies to cross-react with common domains shared between SsZPB2a and sSZPB2c proteins as well as the common ZP domain that is present in all ZP proteins. Please clarify how validation to ensure that only SsZPB2a was the lone target of the antiserum and cross-reactivity, either in Western or immunohistochemistry analyses, was limited. This is worrisome since there is only a 5 kDa difference between these two proteins!
Response 10: Thank you for your careful review and questions. We understand your opinion and worrisome. In our opinion, the antiserum of SsZPB2a is specific to endogenous SsZPB2a according our results. Firstly, the antiserum is obtained by using the full length ORF of SsZPB2a. SsZPB2c is the most similar ZP member but present only 58% identity with SsZPB2a. The relative low similarity makes that SsZPB2c is hardly recognized by antiserum of SsZPB2a. Secondly, WB assay of ovarian tissue using the antiserum of SsZPB2a showed only one clear band, which was around 40 kDa in size, whereas the predicted size of the SsZPB2c protein was 48 kDa. Therefore, the band would not be a manifestation of endogenous SsZPB2c. In summary, we concluded that the antiserum of SsZPB2a was able to specifically bind to the SsZPB2a protein within the ovary.
Point 11: Line 168: The Western blot uses beta-actin for loading control. However, you can see through immunohistochemistry, Figure 2B, that SsZPB2a is largely a secreted protein that accumulates in the extracellular matrix. Therefore, I am not sure that I can agree with the statement, as it reads, in line 156 to 157.
Response 11: Thank you for your careful review and question. SsZPB2a is indeed a secreted protein according to immunohistochemistry, but the protein is expressed in the cytoplasm of oocytes at early developmetnal stages and then accumulated around the mature oocytes. Therefore, the whole proteins from ovarian cells were extracted in this study, and it is accurate that we used beta-actin antibody, which is applicable at the whole cell level in ovary and other tissues, as an internal reference just like other literatures (reference [1][2][3] in this file).
Point 12: Line 177: rTRX is never defined, that I can find, in the entire manuscript.
Response 12: Thank you for your comment. We are sorry for our lack of clarity about this word. We have added the definition information into the revised result 2.2 (Line 184) when “rTRX” first appeared. Its expression is described in the revised Materials and Methods 4.3 part (Line 492-494).
Point 13: Line 225: You state that the red fluorescence intensity is “significantly” lower, as depicted in Figure 7D. I can see that there is lower intensity with 493-siRNA, but not 987-siRNA in Figure 7D. Also, I would refrain from using significant since this wasn’t quantified.
Response 13: Thank you for your question. We apologize for making you misunderstand due to using the wrong image to correspond to the results. In fact, the fluorescence intensity is quantified in Figure 7C, and the Figure 7D showed the histological observation of ovarian tissue after knocking down Sszpb2a. We have used the corresponding picture to explain the results (Line 302 and Line 306). Besides, we have also adjusted the figure number and figure lengend information for figure 7 (Line 316).
Point 14: Discussion: Several grammatical errors throughout the section. Line 255-256: reword sentence
Response 14: Thank you for your suggestion. We have reworded this sentence as “In teleost especially some non-model species, the functions of ZP proteins were not well understood due to the restricted gene editing technology.” (Line 342-344).
Point 15: Line 261: what is ovarian “part”?
Response 15: Thank you for your question. We have changed “ovarian part” to “ovary” in the revised manuscript (Line 349).
Point 16: Line 265-266: reword sentence for clarity
Response 16: Thank you for your suggestion. We have reword this sentence as “ZP protein continues to synthesize until the egg matures in mammals” (Line 353-354)
Point 17: Line 267: “Consistently” ??
Response 17: Thank you for your question. We have changed “Consistently” to “In this study” (Line 354).
Point 18: Line 340: the antibody for Western is stated before the description for how it was produced is introduced. I would suggest moving section 4.5 (that describes the production of the antiserum) to after 4.2.
Response 18: Thank you for your advance. We have moved section 4.5 to after section 4.2 in the revised manuscript.
Point 19: Line 398: “Concentrated” recombinant protein was used to bind to the magnetic beads. Be more specific with quantity of recombinant protein that was used.
Response 19: Thank you for your suggestion. We have used the specific quantity of recombinant protein in the revised manuscript (Line 634).
Point 20: Line 429:“At least number of 100 sperm” to “At least 100 sperm”
Response 20: Thank you for your advance. We have changed “At least number of 100 sperm” to “At least 100 sperm” in the revised manuscript (Line 666).
Point 21: Line 438: “Furtherly” to further
Response 21: Thank you for your suggestion. We have changed “furtherly” to “further” in the revised manuscript (Line 675).
Point 22: Line 442: Specify number, if even 1, that was used for ovarian tissue culture
Response 22: Thank you for comment. We have added the number information for ovarian tissue culture in the revised manuscript (“Three healthy female black rockfish were sampled in February and their ovaries were carefully sampled after being anaesthetized.” Line 685-686).
Point 23: Line 445: Reword sentence
Response 23: Thank you for advance. We have reworded this sentence as “After three minutes of stand, the supernatant was passed through 40 μm cell strainer. Sperm were found in the filtrate by microscopic examination” in the revised manuscript (Line 687-688).
Point 24: Line 452: “referred to a previous study” to “as previously described”
Response 24: Thank you for comments. We have changed “referred to a previous study” to “as previously described” in the revised manuscript (Line 694).
Point 25: Line 453: remove the “.” And provide more detail concerning “appropriate amount”.
Response 25: Thank you for your kindful suggestion. We have reworded this sentence as “After 24 h incubation, 1 μg of two different siRNA targeting ZPB2a and 2 μL of Lipofectamine 3000 transfection reagents (Thermo Fisher Scientific, Waltham, USA) were added to the medium in each well of a 24-well plate” (Line 695).
Point 26: Line 469: “cavity” to activity
Response 26: Thank you for your advance. We have changed “cavity” to “activity” in the revised manuscript (Line 711)
Point 27: Line 472: no need for capitalization of “I” in integrity
Response 27: Thank you for your helpful suggestion. We have changed this word to “integrity” in the revised manuscript (Line 714).
Reference:
- An, R.; Wang, X.; Yang, L.; Zhang, J.; Wang, N.; Xu, F.; Hou, Y.; Zhang, H.; Zhang, L., Polystyrene microplastics cause granulosa cells apoptosis and fibrosis in ovary through oxidative stress in rats. Toxicology 2021, 449, 152665.
- Ghaemimanesh, F.; Ahmadian, G.; Talebi, S.; Zarnani, A. H.; Behmanesh, M.; Hemmati, S.; Hadavi, R.; Jeddi-Tehrani, M.; Farzi, M.; Akhondi, M. M.; Rabbani, H., The effect of sortilin silencing on ovarian carcinoma cells. Avicenna J Med Biotechnol 2014, 6, (3), 169-77.
- Li, Q.; Zhang, Z.; Fan, W.; Huang, Y.; Niu, J.; Luo, G.; Liu, X.; Huang, Y.; Jian, J., LECT2 Protects Nile Tilapia (Oreochromis niloticus) Against Streptococcus agalatiae Infection. Front Immunol 2021, 12, 667781.

Reviewer 2 Report
The authors Rui Li et al. have submitted a manuscript in International Journal of Molecular Sciences (ID: ijms-1860984), entitled “Expression analysis of ZPB2a and its regulatory role in sperm binding in viviparous teleost black rockfish)”.
The authors report the identification and expression of two proteins of the ZP family in the teleost black scorpionfish. These proteins are involved in fertilization between egg and spermatozoa; The number of ZP proteins in particular in teleost’s is greater than in other organisms, because a third round of DNA duplication resulted in the formation of new genes by increasing new functions with neo-sub genic processes. Furthermore, the authors analyzed their temporal and spatial expression profiles for SsZPB2a and SsZPB2c.
Interestingly, assays using ovarian cell culture at which SsZPB2a express is able to influence gene expressions and ability to bind to the sperm head region and delay sperm death.
I believe that all the data of this study can be useful to deepen the knowledge of the ZP family;
Finally, I believe the paper is suitable for publication in the International Journal of Molecular Sciences, after a small revision.
Author Response
Response to Reviewer 2 Comments
The authors report the identification and expression of two proteins of the ZP family in the teleost black scorpionfish. These proteins are involved in fertilization between egg and spermatozoa; The number of ZP proteins in particular in teleost’s is greater than in other organisms, because a third round of DNA duplication resulted in the formation of new genes by increasing new functions with neo-sub genic processes. Furthermore, the authors analyzed their temporal and spatial expression profiles for SsZPB2a and SsZPB2c.
Interestingly, assays using ovarian cell culture at which SsZPB2a express is able to influence gene expressions and ability to bind to the sperm head region and delay sperm death.
I believe that all the data of this study can be useful to deepen the knowledge of the ZP family;
Finally, I believe the paper is suitable for publication in the International Journal of Molecular Sciences, after a small revision.
Response: Thank you for your careful review and suggestion. We have checked and revised the language problems though the manuscript to make it more suitable for publication.
